# ALGEBRANETS

## ABSTRACT

Neural networks have historically been built layerwise from the set of functions in $f : \mathbb{R}^n \to \mathbb{R}^m$, i.e. with activations and weights/parameters represented by real numbers, $\mathbb{R}$. Our work considers a richer set of objects for activations and weights, and undertakes a comprehensive study of alternative algebras as number representations by studying their performance on two challenging problems: large-scale image classification using the ImageNet dataset and language modeling using the enwiki8 and WikiText-103 datasets. We denote this broader class of models as AlgebraNets. Our findings indicate that the conclusions of prior work, which explored neural networks constructed from $\mathbb{C}$ (complex numbers) and $\mathbb{H}$ (quaternions) on smaller datasets, do not always transfer to these challenging settings. However, our results demonstrate that there are alternative algebras which deliver better parameter and computational efficiency compared with $\mathbb{R}$. We consider $\mathbb{C}$, $\mathbb{H}$, $M_2(\mathbb{R})$ (the set of $2 \times 2$ real-valued matrices), $M_2(\mathbb{C})$, $M_3(\mathbb{R})$, $M_4(\mathbb{R})$, dual numbers and the $\mathbb{R}^3$ cross product. Additionally, we note that multiplication in these algebras has higher compute density than real multiplication, a useful property in situations with inherently limited parameter reuse such as auto-regressive inference and sparse neural networks. We therefore investigate how to induce sparsity within AlgebraNets. We hope that our strong results on large-scale, practical benchmarks will spur further exploration of these unconventional architectures which challenge the default choice of using real numbers for neural network weights and activations.

## 1 INTRODUCTION

Nearly universally, the atomic building blocks of artificial neural networks are scalar real-valued weights and scalar real-valued neuron activations that interact using standard rules of multiplication and addition.

We propose AlgebraNets, where we replace the commonly used real-valued algebra with other associative algebras. Briefly, this amounts to replacing scalars by tuples and real multiplication by a tuple multiplication rule. For example, by replacing each scalar weight and activation with $2 \times 2$ matrices, and standard real addition / multiplication with matrix addition / multiplication. These alternative algebras provide three clear benefits for deep learning at scale:

**Parameter efficiency.** One sweeping benefit of AlgebraNets is they are able to match baseline performance on a variety of tasks, spread over multiple domains, with fewer parameters than the competitive real-valued baselines. This means that equivalently capable models can be trained on smaller hardware, and for a given amount of memory, a model with greater effective capacity can be trained. We find some variants of AlgebraNets that are more parameter efficient than the previously considered $\mathbb{C}$ and $\mathbb{H}$ algebras. Throughout the text, we count parameters as the total number of real values e.g. a complex number counts as two parameters.

**Computational efficiency.** For scaling large models, parameter efficiency is not the only bottleneck: FLOP efficiency – reducing the relative number of floating-point operations to achieve an equivalent accuracy – is also important. We find instantiations of AlgebraNets that are more FLOP efficient than the previously considered $\mathbb{C}$ and $\mathbb{H}$ algebras and as FLOP efficient as $\mathbb{R}$. Additionally, all of the proposed algebras offer *parameter reuse* greater than 1 (see Table 1). That is, the ratio of multiplications performed to values consumed is greater than or equal to 1:1. By contrast, for multiplication in $\mathbb{R}$ it is only 1:2. Modern hardware requires a high ratio of floating point operations to bytes loaded (bandwidth) to become compute bound and saturate the arithmetic units. This is

particularly problematic for auto-regressive inference (dominated by matrix-vector multiplies), sparse models, depthwise convolutions and other operations with low arithmetic density.

**Architectural exploration.** The choice of real numbers for weights and activations is usually taken for granted (with some exceptions, e.g. those discussed in Sec. 3). With AlgebraNets, we challenge this established design choice and open up a vast new space for neural network architecture exploration by showing that real numbers can be easily replaced with a variety of algebraic structures. Leveraging these new building blocks, one can consider different algebraic interactions, different choices of non-linearities, and different network architecture choices. Importantly, as we demonstrate in this work, AlgebraNets are not only scalable to large models and complex tasks, but they in fact offer improvements in model efficiency, which makes them a viable practical choice. We believe we have only begun to scratch the surface of what these alternative building blocks can enable, and we hope that their broader adoption will usher in further progress across the field.

In summary, our main contributions are as follows:

- We propose AlgebraNets — a novel class of neural networks, which replaces the nearly ubiquitously used real algebra with alternatives. We show that in contrast to previous work, algebra specific initializations and replacement of batch normalization by an expensive whitening procedure (Trabelsi et al., 2018; Gaudet and Maida, 2018; Wu et al., 2020; Pan et al., 2019) is not necessary, making them a near drop-in replacement to real-valued networks.
- We evaluate AlgebraNets based on a wide range of algebras on three challenging large scale benchmarks: ImageNet image classification (Russakovsky et al., 2015), Enwik8 (LLC, 2009), and WikiText language modelling (Merity et al., 2016).
- We explore sparse AlgebraNets to take advantage of their higher compute density.
- We find that AlgebraNets offer improved parameter efficiency and FLOP parity compared to the real-valued baselines, which establishes them as a viable choice for efficient deep learning at scale.

## 2    AlgebraNets

### 2.1    Why Algebras?

We consider algebras because they have the right properties to make them a drop-in replacement for real numbers in typical neural networks. This is not surprising as the real numbers are an algebra over themselves. An algebra $A$ over a field $K$ (which we take to always be the field of real or complex numbers) satisfies the following properties[1] (Wikipedia contributors, 2020b;a):

1. It is a vector space over $K$.
   - It has an associative and commutative addition operator with an identity element $(\mathbf{x} + \mathbf{0} = \mathbf{x})$ and inverse element $(\mathbf{x} + (-\mathbf{x}) = \mathbf{0})$.
   - It is possible to multiply elements of field $K$ with vectors.[2]
2. There is a right and left distributive multiplication operator $\cdot$ over vectors closed in A.
3. Scalar multiplication combines with $\cdot$ in a compatible way: $(a\mathbf{x}) \cdot (b\mathbf{y}) = (ab)(\mathbf{x} \cdot \mathbf{y})$.

We do not claim that these properties are all *required* as neural network building-blocks, merely that they are convenient. For example, one could imagine not having associative addition – this would require a careful implementation to get right but is possible. One could eliminate the requirement that scalars from $K$ multiply with vectors from $A$ - this would make various normalizations (e.g. batch normalization) impossible, but they are not required. Most importantly, removing some of these requirements does not lead to an obviously useful class of mathematical objects to consider.

In addition to the previously considered $\mathbb{C}$ and $\mathbb{H}$ algebras, we also consider the algebras of $n \times n$ matrices over $\mathbb{R}$ and $\mathbb{C}$ (i.e. $M_n(\mathbb{R})$ or $M_n(\mathbb{C})$) as they have higher compute density than $\mathbb{R}$ and map well to the matrix multiplication units that are becoming common in processors (Oh, 2019). We note

---

[1]We use the terminology 'vector' in the definition as that is the generally accepted mathematical term, however throughout the rest of the paper we use the term 'tuple' instead. This is to avoid the confusion of calling a matrix a vector, which is technically correct in this context, but rife with potential for confusion.

[2]$a(b\mathbf{x}) = (ab)\mathbf{x}$; $1\mathbf{x} = \mathbf{x}$ for 1, the multiplicative identity in $K$; $a(\mathbf{x} + \mathbf{y}) = a\mathbf{x} + a\mathbf{y}$; $(a+b)\mathbf{x} = a\mathbf{x} + b\mathbf{x}$

that $M_2(\mathbb{R})$ is isomorphic to the split-quaternion (Cockle, 1849) algebra and $M_2(\mathbb{C})$ is isomorphic to the biquaternion (Hamilton, 1844) algebra, but the matrix algebras are more familiar so we retain that terminology. Lastly, we consider the dual numbers and the cross product of length-3 tuples.

## 2.2 ALGEBRA DETAILS

| Algebra | Weight Size | Weight Reuse | Multiplies : Values Loaded |
|---|---|---|---|
| Real Numbers ($\mathbb{R}$) | 1 | 1 | 1 : 2 |
| Complex Numbers ($\mathbb{C}$) | 2 | 2 | 4 : 4 |
| $2 \times 2$ Matrix $M_2(\mathbb{R})$ | 4 | 2 | 8 : 8 |
| $3 \times 3$ Matrix $M_3(\mathbb{R})$ | 9 | 3 | 27 : 18 |
| $4 \times 4$ Matrix $M_4(\mathbb{R})$ | 16 | 4 | 64 : 32 |
| $2 \times 2$ Complex Matrix $M_2(\mathbb{C})$ | 8 | 4 | 32 : 16 |
| Quaternions $\mathbb{H}$ | 4 | 4 | 16 : 8 |
| Diagonal Algebras | $N$ | 1 | $N : 2N$ |
| Dual Numbers | 2 | - | 3 : 4 |
| $\mathbb{R}^3$ Cross Product | 3 | 2 | 6 : 6 |

Table 1: Comparison of the different algebras we consider. Weight Size represents the number of components in each tuple, for example a $2 \times 2$ matrix has 4 components. Weight Reuse denotes how many output tuple components each input tuple component is involved in. For example, in a $M_2(\mathbb{R})$ matrix multiply, each weight component is involved in the update of two components.

$\mathbb{R}$; **Real Numbers** All baseline networks are real-valued with scalar weights; standard multiplication rules apply. For two weight values, we load 2 scalars and perform 1 multiply.

We provide tables describing the multiplicative interaction between tuples. The interaction between two tuples $(o_a, o_b, ...) = (t_a, t_b, ...) \bullet (v_a, v_b, ...)$ is described by a matrix where the indices of $t$ are on the left, $v$ are on the top and entries correspond to which component of $o$ the interaction contributes to. A 0 means there is no interaction and a negative index means the result is subtracted.

$$\begin{array}{c|cc} \mathbb{C} & a & b \\ \hline a & a & b \\ b & b & -a \end{array}$$

$\mathbb{C}$; **Complex Numbers** Each weight, $\boldsymbol{w}$, is a length 2 tuple $(t_a, t_b)$ representing the complex number $t_a + t_b i$. For two weight values we load 4 scalars and perform 4 multiplies.

$\boldsymbol{M_n(\mathbb{R})}$; $\boldsymbol{n \times n}$ **Matrices** Each weight is a length $n^2$ tuple, representing an $n \times n$ matrix. Multiplication and addition proceed with standard rules for matrices. We consider up to $M_4(\mathbb{R})$ matrices. For two weight values we load $2n^2$ scalars and perform $n^3$ multiplies.

$\boldsymbol{M_n(\mathbb{C})}$; $\boldsymbol{n \times n}$ **Complex Matrices** Weights are length $2n^2$ tuples representing $n \times n$ complex-valued matrices. We consider only $n = 2$. For two weight values we load $4n^2$ scalars and perform $4n^3$ multiplies. The multiplication table is in Appendix A.

$$\begin{array}{c|cccc} \mathbb{H} & a & b & c & d \\ \hline a & a & b & c & d \\ b & b & \text{-}a & d & \text{-}c \\ c & c & \text{-}d & \text{-}a & b \\ d & d & c & \text{-}b & \text{-}a \end{array}$$

$\mathbb{H}$; **Quaternions** Each weight, $w_i$ is replaced by a length 4 tuple, $(t_a, t_b, t_c, t_d)$. Multiplication is not commutative, with the product of two quaternions given by the Hamilton product (Hamilton, 1843). For two weight values, we load 8 elements and perform 16 multiplies.

**Diagonal Algebra** The high FLOP cost of the whitening operation required by (Trabelsi et al., 2018; Gaudet and Maida, 2018; Wu et al., 2020; Pan et al., 2019) makes networks using it inefficient at training and inference in terms of FLOPs. We attempt to design an algebra where using whitening would in fact be competitive by eliminating the interaction of terms through the algebra. Only when combining the 'diagonal' $\mathbb{D}$ algebra with whitening are there interactions between the different tuple components.

$$\begin{array}{c|cccc} \mathbb{D} & a & b & c & d \\ \hline a & a & 0 & 0 & 0 \\ b & 0 & b & 0 & 0 \\ c & 0 & 0 & c & 0 \\ d & 0 & 0 & 0 & d \end{array}$$

**Dual Numbers** Each weight is represented by a length 2 tuple representing the dual number $(t_0 + t_1 \epsilon)$.

$$\begin{array}{c|cc} & a & b \\ \hline a & a & b \\ b & b & 0 \end{array}$$

For a multiplication, we load 4 values and perform 3 multiplies.

$\mathbb{R}^3$ **Cross Product** Each weight is represented by a length 3 tuple. We use the cross product between two tuples for the multiplication rule, resulting in 6 different multiplies for 6 values loaded.

| $\mathbb{R}^3$ | $a$ | $b$ | $c$ |
|---|---|---|---|
| $a$ | 0 | c | -b |
| $b$ | -c | 0 | a |
| $c$ | b | -a | 0 |

### 2.3 Initialization, Normalization, Non-Linearities, and Pruning

Prior work (Trabelsi et al., 2018; Gaudet and Maida, 2018) has advocated algebra-specific initializations and expensive whitening procedures to replace batch normalization. We find that this is not necessary to achieve good performance, and we are able to use the same initialization, normalization, and non-linearities across all algebras which facilitates exploring a wide variety of options.

To initialize all the components of the algebra tuple at the beginning of a network we set the first tuple component to the typical input. For ResNet, MobileNet, and the RNN we initialize the other components of the tuple with a small one or two-layer MLP, i.e. $t_{b,c,...} = MLP(t_a)$. For the transformer, we take advantage of the fact that the embedding is already a learned representation and simply reshape the output embedding appropriately. We find that the specifics of the input initialization do not have a large effect on performance, though allowing a learnable transformation outperformed initializing additional components to $\mathbf{0}$ or replicating the input. We use standard Glorot (Glorot and Bengio, 2010) weight initialization of each component independently. Comparisons with the algebra specific initializations (Trabelsi et al., 2018; Gaudet and Maida, 2018) can be found in Appendix B.

Existing activation functions can be applied component-wise ($\mathbf{t} = (f(t_a), \cdots, f(t_d))$) and we found that ReLU and swish (Ramachandran et al., 2017) work well; `tanh` and sigmoid can also be applied component-wise as part of GRUs and LSTMs. Applying the activation function to the entire tuple has possible computational advantages if it is ReLU-like as it would allow an entire tuple multiplication to be skipped. For example, consider $\mathbf{t} = f(g(\mathbf{t}))\mathbf{t}$. If $g(\cdot)$ returns the mean of the tuple, and if $f$ was $H$ the Heaviside step function, then one can remove entire components. Appendix B examines different choices for doing this, but we do not consider it further in the main text.

The final logits of an AlgebraNet must be real-valued. We use an Algebra-specific final linear layer and convert the final algebra tuple to a scalar with the tuple-wise $L_2$ norm before applying softmax. More details are in Appendix B.

To apply magnitude pruning (Zhu and Gupta, 2017; Gale et al., 2019) to prune tuples we used the tuple $L_2$ norm as the criterion for pruning for all AlgebraNet variants. For the $M_n(\mathbb{R})$ algebras we also experimented with criteria based on the eigenvalues, $\lambda_i$, and singular values, $\sigma_i$, of each $n \times n$ matrix. The Frobenius norm corresponds to $(\sum_i \sigma_i^2)^{1/2}$ and the determinant corresponds to $(\prod_i \lambda_i)$. We found pruning based on the Frobenius norm to be the most effective, followed by pruning based on the largest eigenvalue. See Appendix C for a comparison between different methods.

(Trabelsi et al., 2018), (Gaudet and Maida, 2018), and (Wu et al., 2020) use whitening in place of batch normalization. Whitening normalizes and de-correlates the different tuple elements from one another. However, this extension results in a substantial increase in both training and test time computational costs, as described in (Pan et al., 2019). The inclusion of the whitening cost to the FLOP count in Fig. 1 highlights the substantial cost inference cost. Cholesky decomposition (Press et al., 2007) of the inverted covariance matrix is required during training and at inference it is not possible to fold the whitening transformation into adjacent convolutions. A contribution from each of the algebra elements contributes to each element in the whitened output. We find that batch normalization does not substantially decrease performance, trains $1.9\times$ faster and has no inference cost, so we use it for all experiments, unless explicitly stated.

## 3 Related Work

(Trabelsi et al., 2018) applied complex-valued networks to convolutional neural networks trained on CIFAR-10, as well as to music transcription and Speech Spectrum Prediction. They find that complex-valued networks with the same number of parameters and more FLOPs perform *slightly* better than real-valued networks. (Gaudet and Maida, 2018) extend the procedure from (Trabelsi et al., 2018) to quaternion valued weights, showing that they are able to reduce the parameter count by a factor of two over complex-valued networks and a factor of four over real-valued networks, while again slightly increasing the top-1 accuracy on CIFAR-10. (Wu et al., 2020) further extend

this approach to octonions (which are a non-associative algebra), demonstrating that they are able to further reduce the parameter count while increasing the accuracy of their models on CIFAR-10.

These papers establish the efficacy of some alternative algebras, though they focus purely on parameter efficiency, rather than FLOP efficiency which is equally important for image classification tasks. Additionally, the tested datasets are relatively small, and it is unclear how the results scale to larger datasets. Both the quaternion and octonion network papers do not test their models on language modeling tasks where parameter efficiency is often of greater importance.

(Parcollet et al., 2018) propose a quaternion recurrent neural network (QRNN) and quaternion LSTM (QLSTM). They show that quaternion based methods are able to reduce the parameter count while offering better performance on the TIMIT and WSJ phoneme recognition tasks. Associative Long Short-Term Memory leverage complex-vectors to increase the memory capacity of LSTMs without a parameter increase (Danihelka et al., 2016).

Recently, many methods to induce sparsity in neural networks have shown that it is possible to train models with an overwhelming fraction of the weights being 0 (Molchanov et al., 2017; Gale et al., 2019; Frankle and Carbin, 2019; Louizos et al., 2018; Evci et al., 2019; Zhu and Gupta, 2017). Many of these methods gradually decrease the number of weights in the network through training by using some combination of each weight's gradient and magnitude. Fine grained sparsity is hard to accelerate on modern hardware, although there have been some recent results demonstrating that speedups are possible (Elsen et al., 2020). (Vecchi et al., 2019) considered inducing sparsity in quaternion networks. Primitives that increase computational density of fundamental interactions would increase the performance of sparse methods as demonstrated on the GPU by (Mueller-Roemer et al., 2019) in scientific computing.

(Jayakumar et al., 2020) emphasize the importance of multiplicative interaction layers providing a particularly useful inductive bias during the fusion of multiple information streams. Specific AlgebraNets may provide strong, useful domain-specific inductive biases, for example as done by (Worrall et al., 2017), leveraging the rotational invariance of complex numbers in convolutional networks and by (Hinton et al., 2018) where they use a $4 \times 4$ pose matrix to represent orientations.

## 4 EXPERIMENTS AND RESULTS

### 4.1 IMAGENET

We examine the performance of AlgebraNet versions of ResNet-50 (He et al., 2016) and MobileNet-v1 (Howard et al., 2017) on the ImageNet (Russakovsky et al., 2015) dataset. We use a width multiplier on the channels to adjust model capacity. For all experiments we use SGD with momentum of $0.9$. We increase the number of training epochs by a factor of two to 180, which we also use for the pruning experiments. This did not affect the baseline, but it improves the pruning results. It also resulted in improved performance for $\mathbb{H}$, so we used it throughout. For a batch size of 256, the initial learning rate for the ResNet experiments was set to $2.5$ and multiplied by 0.1 at epochs 60, 100, 140, and 160. We find it is useful to reduce the amount of $L_2$ regularization that is used for AlgebraNets. The baseline value $10^{-4}$ was reduced by a factor of $0.725$ for ResNet-50 and $0.625$ for MobileNet-v1. We use the swish activation function for all experiments shown in Figures 1 and 2 including the baselines. We found it improved performance across the board.

Figures 1 and 2 compare the trade-offs between accuracy, parameters, and FLOPs for different flavours of AlgebraNet. Notably, we do not find that the parameter reduction without accuracy loss from (Trabelsi et al., 2018) and (Gaudet and Maida, 2018) on CIFAR translates to ImageNet; we are unable to divide the number of parameters by a factor of two/four for $\mathbb{C}/\mathbb{H}$ and match baseline performance. We hypothesize that this is in part due to over-paramaterization of many networks trained on CIFAR and that AlgebraNets act as an additional regularizer, in part due to the greater weight reuse: each tuple component is now involved in multiple equations. We feel that this highlights the need for testing methods on large-scale datasets.

We find $M_2(\mathbb{R})$ AlgebraNets provide the best parameter efficiency of all considered algebras while requiring no more FLOPs than the real baseline on both ResNet-50 and MobileNet-v1. We also find, for both ResNet-50 and MobileNet-v1, $M_2(\mathbb{C})$ AlgebraNets provide better FLOP efficiency than the previously studied $\mathbb{H}$ while having the same ratio of multiplies to values. The diagonal

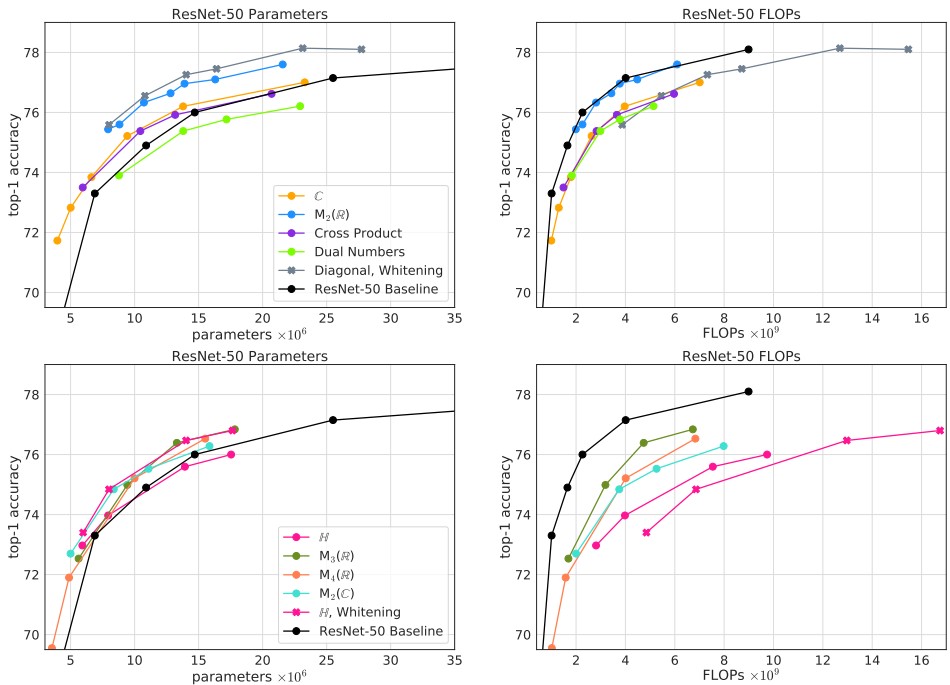

Figure 1: **ResNet-50** We vary the width of a ResNet-50 trained on ImageNet for various flavors of AlgebraNet and the real-valued baseline. We separate based on algebras with up to 1:1 multiplies to values loaded compute density (**top**), and greater than a 1:1 density (**bottom**). In the **left** columns, we show parameters and ImageNet top-1 accruacy. We count parameters as the total number of real values e.g. a complex number counts as two parameters, $M_2(\mathbb{R})$ and $\mathbb{H}$ both count as 4, etc. All runs use batch norm (Ioffe and Szegedy, 2015), unless explicitly stated. **Right:** For the same algebras, we compare the number of floating-point operations (multiply-adds) required at inference. Unlike all previously considered algebras, for $M_2(\mathbb{R})$, we find equivalent computational costs compared to real-valued networks at baseline performance. $M_2(\mathbb{C})$ improves performance compared to the previously considered $\mathbb{H}$ with the same compute density.

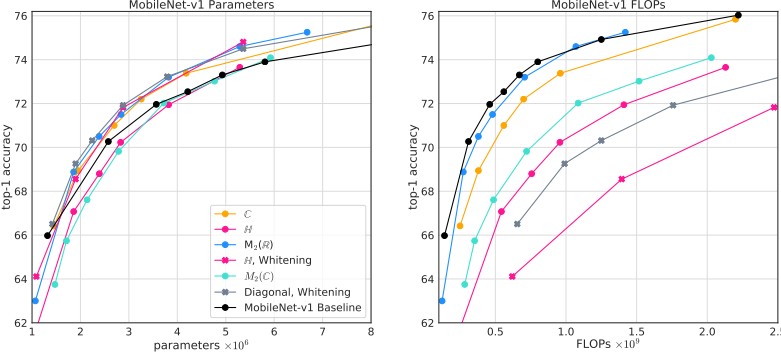

Figure 2: **MobileNet-v1, Left:** We vary the width of a MobileNet-v1 trained on ImageNet for a subset of the considered AlgebraNets and the real-valued baseline. **Right:** For the same algebras, we compare the number of FLOPs (multiply-adds) required at inference.

algebra is extremely parameter efficient and we do find the interaction between different components through whitening to be important as hypothesized – a network trained with whitening achieves 6% higher top-1 accuracy than the same network trained with batch normalization. Unfortunately, adding whitening increases the total number of inference FLOPs by a factor of $3\times$. We are left to conclude that whitening is not currently competitive and recommend using batch normalization for all algebras. Future work exploring the role of the interaction from whitening, and alternatives that are more computationally efficient is an interesting direction. An additional benefit is that AlgebraNets applied

to MobileNet like architectures increase the computational density of the often bandwidth bound depthwise convolutions, while reducing the number of FLOPs in the more costly $1 \times 1$ convolution.

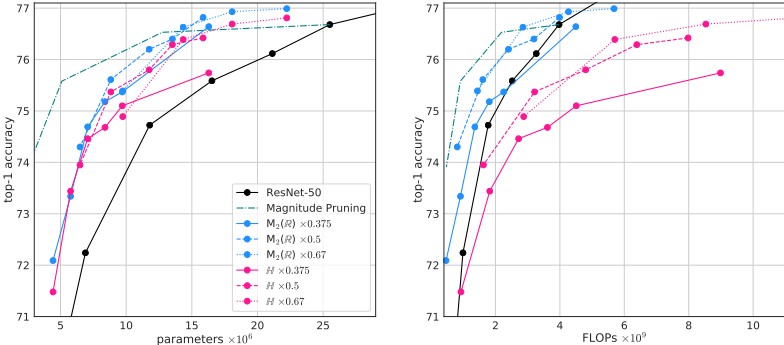

Figure 3: **Pruning ResNet-50, Left:** Using magnitude pruning, we prune entire tuples of $M_2(\mathbb{R})$-ResNet50 to between 50 and 90% sparsity. Different widths correspond to different curves, points along each curve are different sparsity levels. In green, we show baseline magnitude-pruning results from (Gale et al., 2019). **Right:** For the same pruned networks, we show the FLOP efficiency.

## 4.2 Pruning ResNet-50

We use magnitude based pruning, with the schedule proposed in (Zhu and Gupta, 2017). We always begin pruning at 20% and end pruning at 80% of the total training iterations. We prune every 100 steps. At each pruning step, we set tuples with the lowest magnitude, given by $\sqrt{\sum \mathbf{t}_i^2}$, to **0**. We do not prune terms in the final linear layer, the tuple-initialization convolutions, or in the initial convolution of the ResNet. To allow for comparisons with (Gale et al., 2019), we use ReLU activations in pruning experiments, as opposed to swish as used in Fig. 1. Final top-1 accuracies of pruned networks are shown in Fig. 3. Despite pruning entire tuples, which allows skipping an entire tuple multiplication, we are still able to find sparse networks that are similarly FLOP efficient to those from (Gale et al., 2019), while having higher compute density due to the algebra structure. Pruning individual components, rather than setting entire tuples to **0** does improve performance, though does not provide the same computational benefits offered by AlgebraNets. We provide further results in Appendix C.

## 4.3 Transformer-XL on Enwik8 and RNNs on WikiText-103

We perform character level language modeling on the Enwik8 dataset from the Hutter Prize (LLC, 2009) with the Transformer-XL (Dai et al., 2019) architecture. We tuned the baseline model by halving the embedding size, number of heads and feedforward size, resulting in a more challenging 'efficient' baseline with only 25% as many parameters as the 24 layer network from (Dai et al., 2019). We train with Adam (Kingma and Ba, 2014) (learning rate $2 \times 10^{-4}$), dropout 0.25 (component-wise, not tuple-wise), and windows of 1536 at train and 4608 at test time. The $M_2(\mathbb{R})$-Transformer uses a learning rate of .0005 and dropout 0.15. Our 'efficient' baseline 24 layer Transformer-XL model has an embedding size of 512, 4 heads of size 128 each, and a feed-forward hidden size of 1536 for a total

| Name | Algebra | MParams | MFLOPs | Validation BPC |
|------|---------|---------|--------|----------------|
| Transformer-XL 18L (see (Dai et al., 2019)) | $\mathbb{R}$ | 88 | 256 / 426 | 1.03 |
| Transformer-XL 24L (see (Dai et al., 2019)) | $\mathbb{R}$ | 277.4 | 341 / 568 | 0.99 |
| Transformer efficient 24L (ours) | $\mathbb{R}$ | 69.4 | 114 / 227 | 0.99 |
| $M_2(\mathbb{R})$-Transformer efficient 42L (ours) | $M_2(\mathbb{R})$ | 31.2 | 172 / 399 | 0.99 |

Table 2: **Enwik-8**: A parameter-efficient Transformer-XL with performance that matches the classic Transformer-XL on Enwik-8. Switching to $M_2(\mathbb{R})$permits both increasing the number of layers and reducing parameters at the same time - again while keeping the validation bits per character on par. The reported inference FLOPs are representative of an incremental token employing a memory length 1538 or 4608, corresponding to the training and test regimes.

parameter count of 69.4 million. It achieves 0.99 bits per character (BPC), matching the results of (Dai et al., 2019) while requiring 75% less parameters. Using the $M_2(\mathbb{R})$-algebra in all linear layers with fixed activation size results in a further 75% reduction in parameter count. We use these parameter savings to increase the depth of the model from 24 to 42 layers, resulting in a model with 45% as many parameters as the 'efficient' baseline. The resulting $M_2(\mathbb{R})$ AlgebraNet also achieves 0.99 BPC, but with only 31.2 million parameters. The character-embedding layers are computationally unchanged; they associate each character with a $\frac{d}{4}$-sized $M_2(\mathbb{R})$ embedding which can be thought of as a reshaped $\mathbb{R}^d$ embedding. Special consideration has to be paid to the $\mathbb{R}^{l \times l}$ attention matrix, which is often regarded as a practical memory and compute bottleneck (Child et al., 2019; Roy et al., 2020; Kitaev et al., 2020). Using an $l \times l$ algebra valued attention matrix would increase the memory and compute requirements (e.g. by a factor 2 in the Complex-Transformer (Yang et al., 2020) or 4 for $M_2(\mathbb{R})$). Thus, we desire a real valued attention matrix from the sets of $\mathbb{R}^k$-valued key and query vectors $(k, q)$. We do this by reshaping keys and queries from $M_2(\mathbb{R})$ to $\mathbb{R}$. Formally, we redefine the attentions real-valued scalar product as $\langle k, q \rangle_{M_2(\mathbb{R})} := \langle F(k), F(q) \rangle_{\mathbb{R}}$ where $F$ flattens the input into a real vector.

Finally, we also consider a dataset approximately one order of magnitude larger by tokenizing WikiText-103 (Merity et al., 2016) into characters (instead of the more common words). On this dataset we consider a single layer GRU architecture followed by 5 linear readout layers with the ReLU non-linearity, skip connections and layer normalization after each layer. We train using a batch size of 16, Adam (learning rate of $10^{-4}$), and $L_2$ regularization of $10^{-7}$ for 200,000 steps. Training takes two days for the largest baseline variant on a single V100 GPU. We train using length 512 sequences and back propagation through time. We initialize the different components of each algebra with single linear layers from the input. We report results on the typical validation set. We replace a gated recurrent unit (GRU) (Cho et al., 2014) with the AlgebraNet equivalent, as well as replacing the readout layers with AlgebraNet variants. We consider $M_2(\mathbb{R})$, $\mathbb{C}$, $\mathbb{H}$, and $M_3(\mathbb{R})$. Results are shown in Table 3. A $\mathbb{C}$ AlgebraNet with a hidden size of 1024 and 24.1 million parameters achieves a validation BPC of 1.26, comparable to a real-valued network with 1.45 times the parameter count. We find that $M_2(\mathbb{R})$ with a hidden size of 512 results in a validation BPC of 1.30, comparable to a model with twice as many parameters. Again, demonstrating the parameter efficiency of $M_2(\mathbb{R})$ and the usefulness of AlgebraNets for problems such as language modeling where parameter efficiency is crucial.

## 5  CONCLUSION

Conventional neural networks are composed of real-valued weights and activations along with real valued operators. In this work, we proposed AlgebraNets, a general paradigm of replacing real valued weights and operators with weights and operators from other associative algebras in a general fashion. We show these methods to be more parameter efficient than their real-valued counterparts while having higher compute density. We also find that the $M_2(\mathbb{R})$ algebra is more FLOP efficient than previously considered algebras – in fact it is as FLOP efficient as the reals. The increased compute density of the proposed algebras will prove particularly useful for sparse neural networks and auto-regressive inference, due to modern hardware favoring a relatively high compute density. We hope that our work enables further development of these methods and promotes broader research into the fundamental design choices upon which modern neural networks are based.

| Algebra | Size | MParams | MFLOPs | BPC | Algebra | Size | MParams | MFLOPs | BPC |
|---|---|---|---|---|---|---|---|---|---|
| $\mathbb{R}$ | 1024 | 12.1 | 23.6 | 1.34 | $\mathbb{C}$ | 1024 | 24.1 | 80.2 | 1.26 |
| $\mathbb{R}$ | 1280 | 18.7 | 36.7 | 1.32 | $\mathbb{C}$ | 640 | 9.68 | 31.5 | 1.32 |
| $\mathbb{R}$ | 1536 | 27.8 | 52.7 | 1.29 | $M_2(\mathbb{R})$ | 512 | 12.6 | 40.1 | 1.30 |
| $\mathbb{R}$ | 1768 | 35.3 | 69.7 | 1.25 | $\mathbb{H}$ | 512 | 12.6 | 73.6 | 1.32 |
| $\mathbb{R}$ | 2048 | 47.2 | 93.3 | 1.24 | $M_3(\mathbb{R})$ | 384 | 16.4 | 71.9 | 1.31 |

Table 3: **WikiText-103**: We replace a real-valued GRU (and the corresponding linear layers) with AlgebraNet counterparts. We report the minimum validation loss over the last 5% of training. The hidden size is reported as number of tuples, e.g. $M_2(\mathbb{R})$ with 512 tuples has 2048 scalars in total.

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

## A  ADDITIONAL ALGEBRA INFORMATION

### A.1  $M_2(\mathbb{R})$ MULTIPLICATION TABLE

Each tuple $(t_a, t_b, t_c, t_d)$ represents a $2 \times 2$ real matrix.

| $M_2(\mathbb{R})$ | $a$ | $b$ | $c$ | $d$ |
|---|---|---|---|---|
| $a$ | $a$ | $b$ | 0 | 0 |
| $b$ | 0 | 0 | $a$ | $b$ |
| $c$ | $c$ | $d$ | 0 | 0 |
| $d$ | 0 | 0 | $c$ | $d$ |

### A.2  $M_2(\mathbb{C})$ MULTIPLICATION TABLE

Each tuple $(t_a, t_b, t_c, t_d, t_e, t_f, t_g, t_h)$ represents the $2 \times 2$ complex matrix: $\begin{bmatrix} t_a + t_b i & t_c + t_d i \\ t_e + t_f i & t_g + t_h i \end{bmatrix}$

| $M_2(\mathbb{C})$ | $a$ | $b$ | $c$ | $d$ | $e$ | $f$ | $g$ | $h$ |
|---|---|---|---|---|---|---|---|---|
| $a$ | $a$ | $b$ | $c$ | $d$ | 0 | 0 | 0 | 0 |
| $b$ | $b$ | -$a$ | $d$ | -$c$ | 0 | 0 | 0 | 0 |
| $c$ | 0 | 0 | 0 | 0 | $a$ | $b$ | $c$ | $d$ |
| $d$ | 0 | 0 | 0 | 0 | $b$ | -$a$ | $d$ | -$c$ |
| $e$ | $e$ | $f$ | $g$ | $h$ | 0 | 0 | 0 | 0 |
| $f$ | $f$ | -$e$ | $h$ | -$g$ | 0 | 0 | 0 | 0 |
| $g$ | 0 | 0 | 0 | 0 | $e$ | $f$ | $g$ | $h$ |
| $h$ | 0 | 0 | 0 | 0 | $f$ | -$e$ | $h$ | -$g$ |

### A.3  DUAL NUMBER MULTIPLICATION TABLE

Each tuple $(t_a, t_b)$ represents the dual number $(t_a + t_b \epsilon)$.

| Dual Number | $a$ | $b$ |
|---|---|---|
| $a$ | $a$ | $b$ |
| $b$ | $b$ | 0 |

### A.4  CROSS PRODUCT MULTIPLICATION TABLE

Multiplicated uses the cross product between length-3 tuples $(t_a, t_b, t_c)$.

| Cross Product | $a$ | $b$ | $c$ |
|---|---|---|---|
| $a$ | 0 | c | -b |
| $b$ | -c | 0 | a |
| $c$ | b | -a | 0 |

### A.5  LINEAR LAYER EXAMPLE

We give a concrete example of replacing a real linear layer with $M_2(\mathbb{R})$-linear layer such that the activation memory is kept identical. Intuitively, this can be thought of as reshaping the $\mathbb{R}^d$ input activations to have shape $M_2(\mathbb{R})^{d/4}$ that is processed by a $f_M : M_2(\mathbb{R})^{d/4} \to M_2(\mathbb{R})^{d/4}$ linear layer resulting in output activations – when flattened – with shape $\mathbb{R}^d$. Each such linear layer $f_M$ requires $\frac{1}{4}$ of the parameters and $\frac{1}{2}$ of the FLOPS compared to a real $\mathbb{R}^d \to \mathbb{R}^d$ linear layer counterpart.

## B  ALGEBRANET CHOICES: ACTIVATIONS, INITIALIZATIONS, ETC

### B.1  TUPLE-WISE NONLINEARITY

We consider equations of the form:

$$\mathbf{t} \leftarrow f(g(\mathbf{t})) * \mathbf{t} \tag{1}$$

We found that if $g$ is the tuple mean, and $f$ is $H$ the Heaviside function, top-1 performance dropped on an $M_2(\mathbb{R})$ ResNet-50 AlgebraNet by 2.97%. While this drop is significant, the resulting activation sparsity might make it a desirable tradeoff in some circumstances. Other methods, such as setting $g$ to be the determinant resulted in greater than a 10% drop in performance.

## B.2 INITIALIZATION

For a ResNet-50 $\mathbb{H}$-AlgebraNet with the standard number of channels divided by 4, we find a top-1 performance of $74.0 \pm 0.14$ using standard initialization and $74.1 \pm 0.15$ using initialization from Gaudet and Maida (2018). These experiments are done using standard batch normalization instead of the more expensive whitening procedure.

## B.3 CONVERSION TO REALS

For all considered algebras, the norm of the tuple is mathematically given by $\sqrt{\sum_i t_i^2}$. It is possible that the optimal choice for converting to the reals would be different in models with very large final layers, such as word based language modeling – which we do not consider.

## C ALGEBRANET PRUNING

## C.1 ALTERNATIVE TUPLE PRUNING OF $M_2(\mathbb{R})$

For $M_2(\mathbb{R})$, we consider a variety of alternative pruning methods to remove entire tuples, based on the two eigenvalues, $\lambda_1$ and $\lambda_2$ and singular values, $\sigma_1, \sigma_2$. Specifically, because our matrices are square but not symmetric, the Forbenius norm is defined based on the singular values which correspond to the squared eigenvalues of $AA^T$, if $A$ is the matrix in question.

- Frobenius Norm: $\left(\sigma_1^2 + \sigma_2^2\right)^{1/2}$

- Determinant: $\lambda_1 \lambda_2$

- Smallest Eigenvalue $\mathtt{min}(|\lambda_1|, |\lambda_2|)$

- Largest Eigenvalue $\mathtt{max}(|\lambda_1|, |\lambda_2|)$

In all cases, we remove tuples with the minimum magnitude of one of those options.

| Sparsity | det | min | max |
|---|---|---|---|
| 50 | -0.27 | -0.76 | -0.02 |
| 70 | -1.26 | -1.71 | -0.57 |
| 90 | -2.69 | -3.558 | -0.73 |

Table 4: For 50%, 70%, and 90% sparsity, we show the performance relative to the Frobenius norm for different magnitude-based tuple pruning criterion.

In Table 4, we show the resulting drop in top-1 accuracy relative to the Frobenius norm at three different sparsities for three alternative pruning methods. In addition to always achieving the best performance, the Frobenius norm has the additional advantage that it is defined for all Algebra variants that was consider, rather than an $M_n(\mathbb{R})$ specific variant, for example.

## C.2 PRUNING COMPONENTS OF $M_2(\mathbb{R})$ AND $\mathbb{H}$

For $M_2(\mathbb{R})$ and $\mathbb{H}$, we also prune individual tuple elements based on element norms. This equally reduces the number of non-zero weights in the network, though it does not result in entire matrix multiplies that can be skipped.

| Sparsity | $M_2(\mathbb{R})$ | $\mathbb{H}$ |
|:---:|:---:|:---:|
| 50 | +0.20% | +0.18% |
| 60 | +0.37% | +0.32% |
| 70 | +0.58% | +0.49% |
| 80 | +0.93% | +0.74 % |
| 90 | +2.13% | +1.64 % |

Table 5: Performance different from pruning components and entire tuples for $M_2(\mathbb{R})$ and $\mathbb{H}$-AlgebraNets. Depending on the size of the network, the difference between the methods varies slightly. The main point is that pruning elements rather than tuples increases performance, more-so for higher sparsities.

In Table 5, we show the resulting increase in top-1 accuracy that results from pruning individual tuple components, rather than entire tuples. However, due to the structure $M_n(\mathbb{R})$ and $\mathbb{H}$ multiplication, setting individual values to 0 does not result in 0 in the output. Therefore, pruning entire tuples provides more useful computational advantages.

## D  ALGEBRANET TESTS ON CIFAR

We use a network structure based on that described in Gaudet and Maida (2018). We begin with the same ResNet structure, with 128, 256, and then 512 channels in each real block. For the $C$ networks, all channel counts are divided by two. For the $M_2(\mathbb{R})$ and $\mathbb{H}$ networks, we assign the initial convolution, before the residual blocks, to have half the original number of channels, all other channel counts are divided by four. Thus, for $\mathbb{H}$ and $M_2(\mathbb{R})$ we have slightly more than 1/4 the parameters. We train with $24 \times 24$ random crops and evaluate on $32 \times 32$ images.

| Algebra | Parameters ($\times 10^6$) | FLOPs ($\times 10^6$) | top-1 |
|:---:|:---:|:---:|:---:|
| $\mathbb{R}$ | 3.64 | 32.8 | 94.2 |
| $\mathbb{C}$ BN | 1.85 | 33.6 | 94.2 |
| $\mathbb{C}$ W | 1.86 | 50.7 | 94.3 |
| $M_2(\mathbb{R})$ | 0.94 | 20.7 | 94.3 |
| $\mathbb{H}$ BN | 0.94 | 41.4 | 93.4 |
| $\mathbb{H}$ W | 0.97 | 72.9 | 94.1 |

Table 6: A comparison of different AlgebraNets on CIFAR-10. BN denotes Batch Normalization, W denotes the use of whitening.

We find we are able to divide the channels in the filter by two and maintain the same performance using complex valued networks. When reducing the parameter count by a factor of ∼four, we find we are able to again match baseline performance with quaternions and $2 \times 2$ matrices. Regularization has non-trivial effect on performance, and by more finely adjusting the $L_2$ loss for the different algebras may result in higher top-1 accuracy. We note that the relative reduction in parameters on CIFAR-10 is not something we are able to replicate on ImageNet. The results from the main text also hold here – $M_2(\mathbb{R})$ is the only algebra that is able to maintain accuracy while having fewer FLOPs than the baseline real network. For these experiments, we used algebra specific weight initializations, though we again verified that this does not seem to have a substantial effect.

## E  EXAMPLE $M_2(\mathbb{R})$ CODE

We write the update rule explicitly for readability. Note that it is possible to concatenate the relevant terms on the channel axis to reduce the number of convolutions needed.

CONVOLUTION

```
"""
Simplified example code for M_2(R).
x: Input with an additional algebra axis. In the case
   of a convolution, either (B, H, W, C, A) or
   (B, C, H, W, A)
w: Corresponding weight matrix, with an additional
```

```
    algebra axis.
"""

# Rule that describes 2x2 matrix multiplication.
mat_22_rule = [[(0, 0), (1, 2)],
               [(0, 1), (1, 3)],
               [(2, 0), (3, 2)],
               [(2, 1), (3, 3)]]
# Update each of the four algebra components.
x_new = [0, 0, 0, 0]
for i in range(4):
    for j in range(2):
        # w: weight with an extra algebra dimension.
        # x: Input with shape [B, ... , A] where A is the additional algebra dimension.
        x_new[i] += Conv2D(x[..., mat_22_rule[i][j][1],
                           w[..., mat_22_rule[i][j][0],
                           ...)
# Add bias if wanted. Add (4,) to shape.
```

LINEAR LAYER

```
# Update each of the four algebra components.
x_new = [0, 0, 0, 0]
for i in range(4):
    for j in range(2):
        # w: weight with an extra algebra dimension.
        # x: Input with shape [B, L, A] where A is the algebra dimension.
        x_new[i] += dot(x[..., mat_22_rule[i][j][1],
                        w[..., mat_22_rule[i][j][0])
# Add bias if wanted. Add (4,) to shape.
```

# F  INCREASED ACTIVATION MEMORY

Due to the activations, there will be a slight increase in memory footprint from AlgebraNets in some cases. For example, in a $M_2(\mathbb{R})$ AlgebraNet for ResNet-50 with channels/4, there will be C/4 convolutions performed. This would, in a naive implementation, result in twice the activation memory. However, with a properly written kernel, this would not be the case. There is, however, an additional factor: to reach comparable performance a slightly larger network than C/4 is needed. In practice about a $1.3\times$ increase in activation memory would be incurred.

