# OpenReview forum: "AlgebraNets"
_ICLR.cc/2021/Conference — Reject_

### Official Review · AnonReviewer2 · 2020-10-25
**Interesting study of replacing the traditional real-valued algebra with other associative algebras**

**Rating:** 6
**Confidence:** 2

**Review:**

The paper proposes an interesting kind of networks, AlgebraNets, which is a general paradigm of replacing the commonly used real-valued algebra with other associative algebras. This paper considers C, H, M2(R) (the set of 2 × 2 real-valued matrices), M2(C), M3(R), M4(R), dual numbers, and the R3 cross product, and investigates the sparsity within AlgebraNets.

The work in the paper is interesting and this paper is generally written well. However, there are a few issues/comments with the work:

1.The citation of the references in the main body of this paper is not easy to read. It will be better to replace the format “author(s) (year)” with the format “(author(s), year)” ;

2.Some figures and tables do not appear near the discussion, for example, Figure 1 is shown on Page but it is discussed until page 5, which makes it difficult to read;

3.In Figure 1, the subfigure in the second row and first column, it seems that the performance of model with H and whitening the best stable performance.  The subfigure in the second row and second column, it can be seen that the model with H  is not better than the baseline model;

4.There are many inconsistencies in the format of the reference, for example,

1)In some places the author's name is abbreviated, while in others it is not. References “C. J. Gaudet and A. S. Maida. Deep quaternion networks. In 2018 International Joint Conference on Neural Networks (IJCNN), pages 1–8, 2018. ” and “Geoffrey E. Hinton, Sara Sabour, and Nicholas Frosst. Matrix capsules with em routing. In ICLR, 2018. ”;

2)In some places the conference’s name is abbreviated with the link, while in others it is not. References “Siddhant M. Jayakumar, Wojciech M. Czarnecki, Jacob Menick, Jonathan Schwarz, Jack Rae, Simon Osindero, Yee Whye Teh, Tim Harley, and Razvan Pascanu. Multiplicative interactions and where to find them. In International Conference on Learning Representations, 2020. URL https://openreview.net/forum? id=rylnK6VtDH.” and “Geoffrey E. Hinton, Sara Sabour, and Nicholas Frosst. Matrix capsules with em routing. In ICLR, 2018. ”.

Please check carefully and correct the inconsistencies.

+++++++++++++++++++++++++++++++++++++++++++++++++++++++++++++++++++++++++++++++++++++++

The paper replaces the traditional real-valued algebra with other associative algebras and shows its parameter and FLOP efficiency.  In the beginning, "I think it is an interesting piece of work, and it may be helpful to develop the basic structural design of neural networks. ". However, after getting the response from the author(s), I more doubt the significance of the work in this paper: although many types of models have been proposed in this paper, the improvement over the baseline models is limited. I did not lower the grade on this paper since I thought it would be interesting and important (if effective) to extend the traditional real number field to more complex algebraic structures.

+++++++++++++++++++++++++++++++++++++++++++++++++++++++++++++++++++++++++++++++++++++++

---

> ### Author Response · Authors · 2020-11-14
> **Thank you for the review**
>
> Thank you for your careful read of our manuscript. We appreciate the comments very much, and will update the paper with the changes to the references and try to better stagger the introduction and reference to the different figures in the manuscript.
>
> Are there technical issues we can address/clarify/improve that would help improve the perception of our work?

---

> > ### Comment · AnonReviewer2 · 2020-11-14
> > **Please clarify my question**
> >
> > Thanks for your response.
> >
> > Sure, there are. Because you don't respond to my above item 3. If the performance could not be improved, what is the meaning of your complex model?

---

> > > ### Author Response · Authors · 2020-11-14
> > > **Response to Item 3**
> > >
> > > Sorry for not responding to the third point-- in Figure 1, we are showing performance per parameter (left) and performance per FLOP (right). This is important because while an algebra may be able to reduce the parameter counts, there may be an increased FLOP cost, especially due to a procedure like whitening. We note that in earlier work (Deep Complex Networks and Deep Quaternion Networks) the cost of whitening was largely ignored, since results focused on parameter efficiency. In terms of performance-per-parameter, many algebras are actually more performant than the baseline. However, in terms of FLOPs, only M_2(R) is able to match the baseline performance. It is important to note, however, that these algebras have added benefits: specifically, the higher compute density. Aside from being important for sparsity, with proper kernels (or even hardware!) the performance gap may be negligible.

---

> ### Author Response · Authors · 2020-11-20
> **Anything more we can address?**
>
> As the discussion period ends soon, we were wondering if there are any more concerns we could address to help increase your score of our work?

---

### Official Review · AnonReviewer4 · 2020-10-28
**Impactful paper with strong empirical results**

**Rating:** 7
**Confidence:** 4

**Review:**

## Summary
The authors propose AlgebraNets - a previously explored approach to replace real-valued algebra in deep learning models with other associative algebras that include 2x2 matrices over real and complex numbers. They provide a comprehensive overview of prior methods in this direction and motivate their work with potential for both parameter and computational efficiency, and suggest that the latter is typically overlooked in prior literature. The paper is very well-written and follows a nice narrative, and the claims are mostly backed empirically with experimental results.
## Pros
* Empirically justified with experiments on state-of-the-art benchmarks in both computer vision and NLP.
* Establishes that exploring other algebras is not just an exercise for mathematical curiosity but also practical, and encourages deep learning practitioners to extend the results.
* Perhaps the most useful aspect is that the experiments fit well into a standard deep learning framework – with conventional operations, initialization, etc. That is, the algebras do not require significant custom ops/modifications to achieve state-of-the-art results.
* Shows multiplicative efficiency (parameter count and FLOPs) in many cases
## Cons
* The authors motivate this work with computational efficiency; however, I did not find any discussion or comments on the total memory footprint. Do any of the algebras require us to keep track of partial computations/intermediate steps - subsequently increasing the total memory footprint? In the case of vision examples, which are dominated by the activations, what are the implications? If the memory footprint is indeed not consistent with a real-valued algebra, then are we trading model/input size for fewer parameters/efficient computation?
* An intuitive justification of the algebras used in these experiments, along with insight for future algebras might be a nice addition, although I wouldn't consider it a con.
* Are certain algebras more amenable to specific hardware architectures? If so, a brief discussion would enhance the paper overall.

---

> ### Comment · AnonReviewer3 · 2020-11-14
> **A concern of any new insights over "Deep Complex Networks" ICLR 2018.**
>
> Dear Reviewer,
>    I have a concern: whether this work provides any new insights over "Deep Complex Networks" ICLR 2018.
> https://openreview.net/forum?id=H1T2hmZAb
>
>    I am not sure whether there is enough value to support this work appear in a top AI conference.  Would like to hear your opinions.

---

> > ### Author Response · Authors · 2020-11-14
> > **We feel there are many new insights and contributions.**
> >
> > Deep Complex Networks is an interesting paper that highlighted some of the potential of investigating these alternate algebras. However, they only investigate a single algebra (complex numbers) and do not recognize the increased compute density of algebra nor explore pruning or sparsity inducing methods that would greatly benefit from this increased compute density on modern hardware. Additionally, while their proposal is parameter efficient, it is not FLOP efficient due to the computationally expensive whitening procedure.
> >
> > We test a large number of algebras and find an algebra (2x2 matrix rings) that actually work better than anything that has previously been looked at, in terms of performance per FLOP. Additionally, we show that we do not need some of the complexities discussed in earlier work exploring these algebras: specific initialisation schemes, for example, do not seem to matter as much.
> >
> > Lastly, we find some crucial differences in terms of the efficacy of these algebras in testing at scale. Using ImageNet instead of CIFAR-10 one does not recover the same performance per-parameter. To further test this regime, we also use the more computationally efficient MobileNet. Finally, we test the most promising algebras on a variety of different domains as well.
> >
> > Deep Complex Networks was an exciting work but we think we have made a series of new contributions that are important to anyone interested in complex networks or other algebras.

---

> > ### Comment · AnonReviewer4 · 2020-11-16
> > **Re: concern of new insights**
> >
> > There seems to be quite ab bit of discussion regarding the title, and if calling it "AlegbraNets" might have overpromised and underdelivered; I understand the other reviewers' concerns. However, upon re-examining the paper, I believe there may be enough merit to warrant acceptance. I agree with other reviewers' that this work may not be entirely novel (which I point out in my review as well); however, I see this is as a valuable contribution for the following reasons:
> >
> > 1. To my knowledge, one of the first publications to empirically show the value of other algebras on established datasets (ImageNet, enwiki-8) and respective near-SOTA model architectures such as transformer-xl. Deep complex networks motivates the line of research, but I would consider CIFAR-10/100 to be more of toy examples. These results, if broadly disseminated, has the potential to encourage subsequent contributions. I see the leap from examples to larger-scale results as impactful. If there are other publications that establish similar results, please share and I can certainly be convinced otherwise.
> >
> > 2. On a similar note, a search (and exploration) of algebras beyond complex numbers is valuable and their recommendation of using 2x2 matrix rings as optimal under a computational notion is promising.
> >
> > 3. Upon seeing the authors' response on memory footprint, I do see there is a tradeoff between the computation and memory footprint, making it a design choice. Something that would strengthen the paper a bit  more is if they can define a cost model based on standard hardware (e.g. GPU/TPU) and show how using a 2x2 matrix algebra is conclusively better than real numbers.
> >
> > Overall, my vote of confidence is for the empirical results on widely adopted convolutional models, transformer-xl etc., the ease of usage, etc. This could be one of the papers that spur the paradigm shift from real numbers to other algebras in the SOTA spectrum of models. Unless there are prior results I am unaware of that have shown similar results

---

> ### Author Response · Authors · 2020-11-14
> **Thank you for your review**
>
> Thanks for the review. We have tried to address your section of cons below, and will update the text in the next few days to reflect these changes. We, of course, thank you for listing the pros of our work, we agree that the exploration of alternate algebras is both useful and impactful!
>
> > The authors motivate this work with computational efficiency; however, I did not find any discussion or comments on the total memory footprint. Do any of the algebras require us to keep track of partial computations/intermediate steps - subsequently increasing the total memory footprint? In the case of vision examples, which are dominated by the activations, what are the implications? If the memory footprint is indeed not consistent with a real-valued algebra, then are we trading model/input size for fewer parameters/efficient computation?
>
> There are two issues that could increase the memory footprint.  The first is that regardless of implementation, the number of activations will be larger by a factor of about 1.3 (empirically) for the M2R networks when matching the performance of the real network.
>
> The second issue is that with our current implementation, there are indeed intermediate feature maps that could increase the memory usage.  For M2R there are 8 convolutions of size C/4, which means the memory usage would approximately be doubled.  However, we note that if the appropriate kernels were written to perform the algebra calculation at the lowest level, then this doubling overhead would not exist.
>
> We will update the text saying that this is a possible concern and point out mitigating strategies.
>
>
> > Are certain algebras more amenable to specific hardware architectures? If so, a brief discussion would enhance the paper overall.
>
>
> The matrix algebras would map nicely to the currently popular systolic arrays common on accelerators such as GPUs and TPUs.  Although the arrays on current GPUs and TPUs are bigger than sizes considered here, it is possible that future hardware could move to smaller arrays.  Having a larger number of smaller systolic arrays would map nicely to sparse algebra networks.  It would also be possible to build specific algebra multipliers at the hardware level for any algebra.
>
> These algebras would also accelerate inference cases that would otherwise have a batch size 1 and be completely bandwidth limited, by increasing the compute density even in this case.
>
> We agree this is an interesting direction and will add a section to the appendix that emphasises this further.

---

### Official Review · AnonReviewer3 · 2020-10-30
**Huge title without convincing contribution**

**Rating:** 5
**Confidence:** 4

**Review:**

In this paper, the authors propose the usage of complex numbers in deep neural networks. Would be good to know that complex numbers, n x n matrices, quaternions, diagonal matrices, etc. all can be used in neural networks. The authors also claims benchmark performance in large-scale image classification and language modeling.

However, this work cannot be appreciated due to the following aspects:
1. A first question is "Why it is necessary?"  Interestingly, the authors already included Section 2.1 Why Algebras?   However,  I am not convinced at all.  A good answer may take either of the two forms: A). simply a math step showing great potential behind;  2) large-scale neural networks that have engineering advantages.  It seems that the authors took the second approach, however, ImageNet is not that challenging and there may be no clear need to switch to complex numbers.  Would the authors be able to justify this?

2. Then, the authors directly go to evaluations. The figures seem to show good advantages.  However, could you please justify your x,y-axis?  The reported results look high biased. As a reviewer, I have to doubt that the authors may have selectively present their results.

    A good research paper on such a big topic, should give clear methodology first, right?  If the methodology is questionable, such good results may become noise to the community.
    I would hope the authors clarify their methodology, and then present that advantages obtained in the experiments.

3. As a top AI conference, I believe that we are looking for intellectual contributions.
    This paper is working on a huge title, which is attractive. However, when I try to identify the intellectual contributions (can be theory, algorithm, engineering, applications), I am not convinced at all.  I know such a topic is not easy to handle. I would simple ask the authors to respond to a direct question: how would like the community to appreciate your work?

NOTE: a lot of disputes are around "the huge title 'AlgebraNets'". However, I did not receive justification response from the authors. A possible reason may be the authors are not aware of how big the topic it is, and were so attractive/confident in the current experimental improvements (which is also very appreciated).

---

> ### Author Response · Authors · 2020-11-14
> **Thank you for your review**
>
> Thank you for your review. Below, we’ve tried to answer your questions, but firstly here is our motivation for this work, which we hope will help frame both the manuscript and our response.
> a) We wanted to search for more efficient alternatives to real numbers to use in neural networks.  This was the goal from the beginning.  We were especially interested if we could combine the higher compute density of algebras with sparsity.
> b) There had been some prior work showing complex numbers and quaternions were more parameter efficient but nothing about FLOPs. FLOPs are correlated with runtime and often at least as important as parameter efficiency, especially in vision models.
> c) We noticed the prior work on complex and quaternions used very FLOP expensive whitening and special initialization.
> d) We chose to investigate those algebras and many more on both a parameter and FLOP efficiency basis. Furthermore, we made preliminary steps towards testing sparsity inducing techniques and these algebras.
>
> In response to your specific queries:
> * Do you mean why are more efficient neural networks necessary?  Or why are different algebras necessary for more efficient networks?  They are one approach to finding more efficient architectures, but certainly not the only one.
> We are surprised that you do not think ImageNet is not a suitable task for demonstrating this method: it is a challenging task and is widely used as a way to benchmark state-of-the-art methods. We also have results on enwik8 and wikitext-103 language modeling which, while certainly not large by GPT-3 standards, has been considered a standard language modeling benchmark in the literature.  What tasks would you like to see?
>
> * Thanks for commenting that the results look promising. We choose our axes based upon the standards in other work on efficiency in neural networks.  For example: EfficientNet (M. Tan, et al 2019) show results as FLOPs/parameters vs top-1 accuracy. Similarly, MobileNet (A.G. Howard et al 2017) and MobileNet v2 (M. Sandler et al 2018) also present the same axes. Many pruning papers also use these same axes, for example, “What is the State of Neural Network Pruning?” from D. Blalock et al 2020 and “The State of Sparsity in Deep Neural Networks” from T. Gale et al 2019.  We thank the reviewer for commenting that they found some of the methodology unclear -- are there certain aspects that you found to be particularly confusing?
>
> * We feel there are a series of important contributions in the work:
>  a.) We find some complexities from prior works are not needed.  For example, we do not need special initializations for good performance.
> b.) Clear demonstration of which algebras are more efficient both in terms of parameters and FLOPs in a modern regime across multiple domains.
> c.) We discover that M_2R is better than all algebras that have been previously considered in terms of performance per FLOP while still offering a substantial parameter reduction.
> d.) Showing that M_2R networks can be made sparse and will be better than normal sparsity due to the higher compute density of the algebra.
>
> We hope that this helps address your concerns.  Please do let us know if there is anything more we can clarify.

---

> > ### Comment · AnonReviewer3 · 2020-11-14
> > **Still huge title without convincing contributions**
> >
> > 1. "Do you mean why are more efficient neural networks necessary? Or why are different algebras necessary for more efficient networks? "
> >      The target project aims to improve efficiency by complex algebras, I am interested why go for it?  yes, there is performance gain, also there is overhead, and not compatible in TensorFlow/PyTorch (as is used by the wide community).  Why it is necessary to go for it?  It is OK to be a small group of researchers or a particular industrial product.   As in the following points, I think claiming ImgeNet should go for it is not convincing enough.
> >
> > 2. Do not be "surprised that you do not think ImageNet is not a suitable task for demonstrating this method: it is a challenging task and is widely used as a way to benchmark state-of-the-art methods. We also have results on enwik8 and wikitext-103 language modeling"
> >      The reasons are we already have very effective neural networks. I do not see clear reason why we urgent engineers switch to much more complex algebras.  If the reviewers vote for an acceptance of "AlgebraNets" to ICLR, some improvements on well-studied datasets are not enough to justify why ICLR accepts such a big title.
> >      The concern is not "What tasks would you like to see?"  but why engineers need such a switch.
> >
> >     Another very direct question would be: the compared schemes in ImageNet were targeting at improving accuracy, now your results claiming better computation efficiency. Actually, more fair comparison would be those compression schemes (EfficientNets, MobileNets (included), complex-valued nets), right?   The current presentation of the evaluation methodology is not convincing.
> >
> > 2. "We feel there are a series of important contributions in the work: a.)   b) c) and d)"
> >      Those claims are interesting, but are far from a support of "AlgebraNets".  The experiments and claims are from two tasks (on two datasets), which are not enough.

---

> > > ### Author Response · Authors · 2020-11-16
> > > **Response**
> > >
> > > The reviewer seems to believe that using the algebras in this paper would be a large engineering challenge.   We note that it is fairly simple to implement these networks in Tensorflow or Pytorch, just as we’ve shown in the appendix for JAX.  This does leave some performance on the table, but this is a natural progression for almost all new ideas. First it is demonstrated that they work, then later maximum performance implementations are created.  The initial implementations of real-valued convolutions were far from optimal as one example.
> > >
> > > It is also odd to claim that “some improvements on well-studied datasets are not enough” when probably a majority of all papers accepted will do exactly this — show improvements on well-studied datasets.  Indeed, one should be skeptical of claims on poorly studied datasets, it is far harder to show gains on well studied datasets.  We strongly disagree with the implication that because "we already have very effective neural networks", research on _more_ effective techniques is not necessary.
> > >
> > > The reviewer repeatedly claims that “ImageNet … is not convincing enough” and “experiments and claims are from two tasks (on two datasets) which are not enough”, but then when we ask directly for which additional tasks would be useful to include, they instead say “The concern is not ‘What tasks would you like to see?’ but why engineers need such a switch.”  If the reviewer could clarify their position, the authors would find it most helpful.  And for what it’s worth, we would like to clarify that we believe we have three tasks and four datasets.  Image classification, character level and word level language modeling are the tasks, and ImageNet, CIFAR-10 (appendix), enwik8 and WikiText-103 are the datasets

---

> > > > ### Comment · AnonReviewer3 · 2020-11-21
> > > > **You do not understand.**
> > > >
> > > > If you do not understand them, try to discuss them in detail with your mentors. It is OK to reply/respond that you have better logics while the reviewer may not fully appreciate your work, but do not try to dispute without fully understanding the comments.

---

> ### Author Response · Authors · 2020-11-21
> **Alternative Title?**
>
> Because the reviewer has such strong concerns about the title, we wonder if changing the title would allow the reviewer to reconsider their opinion on the rest of the paper?

---

> > ### Comment · AnonReviewer3 · 2020-11-21
> > **Would like to reconsider if you can address the following concerns.**
> >
> > 1.  Please point out tasks that may have clear need to switch to the proposed algebra sets.  I am NOT saying that "the provided tasks in the paper are not important". I am questioning that in a top AI conference, what kind of readers would benefit from such an engineering switch, proposed by this paper.
> >
> >      I notice the authors questions the review comments. But actually, the meaning and question is like the above.
> >
> >      I wrote several sentences to encourage the authors to provide responses:
> >
> >     "Would the authors be able to justify this?"  "The concern is not "What tasks would you like to see?" but why engineers need such a switch."
> >
> >     "I would hope the authors clarify their methodology, and then present the advantages obtained in the experiments."
> >
> >     "I would simply ask the authors to respond to a direct question: how would you like the community to appreciate your work?"
> >
> >      "ImageNet is not that challenging and there may be no clear need to switch to complex numbers".  For example, if the authors believe ImageNet is your choice, please give more clear reasons for asking many engineers/readers to try your approach. Please be advised, if ICLR accepts this paper, many readers will be interested and would like to try it out. If you can clearly justify it, I would be happy to re-evaluate.
> >
> >      You previous response "We are surprised that you do not think ImageNet is not a suitable task for demonstrating this method: it is a challenging task and is widely used as a way to benchmark state-of-the-art methods. " does not address my above concern.
> >
> >
> > 2.  A second concern about your experiments "I would hope the authors clarify their methodology, and then present the advantages obtained in the experiments."
> >
> >     You mentioned some recent and close work use similar performance metrics.  However, the question is would you please "clarify the experiment methodology".  Mentioning recent work is good, but does not FULLY address my concern.
> >
> > 3.  Which category would you claim your work to be "Doing something new, doing something important, doing something new and important"?
> >
> > 4. Please try to point out the scientific values behind. It can be simple but effective. For example, when I read the manuscript, one can hardly believe "2x2 matrix rings)... work better than anything ...." this claim is rather problematic.  Try to provide answers and convince the reviewer and future readers, using two or three sentences (clear logics).
> >
> > 5. This exaggerated claim raised concerns about the rigorous of the methodology of this work.
> >
> >     Your response can justify that your work has not such exaggeration.  Try to provide response that address such a concern, if future readers also challenge such a exaggeration.
> >
> >
> > All the above concerns (and some other in my comments), are asking for your clarifications.
> >
> > If you do not understand them, try to discuss them in detail with your mentors.  It is OK to reply/respond that you have better logics while the reviewer may not fully appreciate your work, but do not try to dispute without fully understanding the comments.

---

### Author Response · Authors · 2020-11-16
**Updated Version**

We would like to thank the reviewers for the comments. We have updated the manuscript. We have moved the figures such that they appear closer to where they are referenced. We updated the citation style, as asked for. We also standardised the presentation of citations. In the supplement, we added a discussion of the change in activation memory.

---

### Decision · Program_Chairs · 2021-01-07
**Final Decision**

**Decision:**

Reject

**Comment:**

The paper proposes to deep neural network models with elements of the weight from algebras, and considers a wide range of algebras and large scale promising experiments. The paper raised a heated discussion.

Pros:

- Using algebras, one can hope for more efficient architectures

- Numerical experiments on a wide range of problems

Cons:

 - The theoretical grounding provided in the current version of the paper is not sufficient. The study is empirical (nothing wrong about it), but there is no clear understanding/explanation of why particular choice is better than another, and also why it works in the particular setup.

- The title does not reflect the content of the paper. It is too broad, and also in some sense “provocative”. The reader expects something much more significant from it.

- Experiment setup: the resulting flops/accuracy figure (main result, Figure 1) does not contain error bars.  I.e., the accuracies should be averaged over several random seeds in order to guarantee the resulting metrics. Also, this figure does not show a clear advantage over the ResNet-50 baseline.